# Pancreatic β Cells Inhibit Glucagon Secretion from α Cells: An In Vitro Demonstration of α–β Cell Interaction

**DOI:** 10.3390/nu13072281

**Published:** 2021-06-30

**Authors:** Wenqian Gu, Camilla Christine Bundgaard Anker, Christine Bodelund Christiansen, Tilo Moede, Per-Olof Berggren, Kjeld Hermansen, Søren Gregersen, Per Bendix Jeppesen

**Affiliations:** 1Department of Clinical Medicine, Aarhus University, Palle Juul-Jensens Boulevard 165, 8200 Aarhus N, Denmark; WQG@novozymes.com (W.G.); camilla_anker@me.com (C.C.B.A.); c.christiansen@clin.au.dk (C.B.C.); kjeld.hermansen@clin.au.dk (K.H.); soeren.gregersen@aarhus.rm.dk (S.G.); 2The Rolf Luft Research Center for Diabetes and Endocrinology, Karolinska Institutet, 171 77 Stockholm, Sweden; tilo.moede@ki.se (T.M.); per-olog.berggreen@ki.se (P.-O.B.); 3Department of Endocrinology and Internal Medicine, Aarhus University Hospital, Palle Juul-Jensens, Boulevard 99, 8200 Aarhus N, Denmark; 4Steno Diabetes Center Aarhus, Aarhus University Hospital, Palle Juul-Jensens Boulevard 99, 8200 Aarhus N, Denmark

**Keywords:** MIN-6, α-TC6-1, α–β cell interaction, glucagon secretion, insulin secretion, co-culture

## Abstract

Interactions between endocrine α and β cells are critical to their secretory function in vivo. The interactions are highly regulated, although yet to be fully understood. In this study, we aim to assess the impact of α and β cell co-culture on hormone secretion. Mouse clonal cell lines α-TC6-1 (α cell line) and MIN-6 (β cell line) were cultured independently or in combination in a medium containing 5.5, 11.1, or 25 mM glucose, respectively. After 72 h, hormone release was measured using insulin and glucagon secretion assays, the cell distribution was visualized by inverted microscopy and an immunocytochemistry assay, and changes in gene expressions were assessed using the RT-PCR technique. The co-culture of the two cell lines caused a decrease in glucagon secretion from α-TC1-6 cells, while no effect on insulin secretion from MIN-6 cells was revealed. Both types of cells were randomly scattered throughout the culture flask, unlike in mice islets in vivo where β cells cluster in the core and α cells are localized at the periphery. During the α–β cell co-culture, the gene expression of glucagon (Gcg) decreased significantly. We conclude that islet β cells suppress glucagon secretion from α cells, apparently via direct cell-to-cell contact, of which the molecular mechanism needs further verification.

## 1. Introduction

Islets of Langerhans, anatomically complex micro-organs located in the pancreas, crucially influence glucose homeostasis. The cellular composition and architecture of the islets differ between species [1,2]. Mammalian pancreatic islets consist of several secretory endocrine cell types; the insulin-secreting β cells, the glucagon-producing α cells, the somatostatin-releasing δ cells, the pancreatic polypeptide-producing (PP) cells, and ghrelin-producing ε cells [3,4]. While these cells are randomly scattered throughout human islets, the predominating β cells are clustered in the central core of rodent islets, surrounded by a mantle layer of non-β cells [3,5,6]. The islet cells selectively interact with each other and form a complex regulatory network, which is essential for optimal islet function [7]. The interactions within islets are mediated through paracrine and autocrine signaling as well as via connection molecules between islet cells [8].

Hormones are the primary auto/paracrine signaling molecules within the islet. The presence of glucagon increases both insulin and somatostatin release. Somatostatin is a strong inhibitor of glucagon and insulin secretion, while insulin boosts somatostatin production but exerts an inhibitory action on glucagon release. Furthermore, neurotransmitters and neuropeptides such as glutamate and γ-aminobutyric acid (GABA) also play important roles in auto/paracrine signaling [9,10,11,12,13]. In addition, the connection molecules between endocrine cells are an integral part of cell-to-cell interaction, and include cell adhesion molecules (N-CAM, cadherins, etc.), gap junctions, ephrin receptors, and ligands [8].

Pancreatic α and β cells constitute most of the islet. Human islets are composed of circa 60% β cells and circa 30% α cells [14,15,16]; in mouse islets, >70% of the cells are β cells and <20% are α cells [1,17]. Insulin is secreted by β cells in response to increased blood glucose levels and act on cells throughout the body by stimulating the uptake, utilization, and storage of glucose. The pathophysiology of type 1 diabetes (T1D) is explained unequivocally by severe insulin deficiency, whereas both insulin resistance and insulin deficiency characterize the pathophysiology of type 2 diabetes (T2D). The main action of glucagon produced by α cells is to stimulate glucose production by the liver through glycogenolysis and gluconeogenesis. In T2D, fasting plasma glucagon levels are typically increased by approximately 25% and glucagon levels do not decrease as much as they do in healthy individuals, e.g., following glucose absorption. Fasting plasma glucagon is also elevated in T1D patients and in streptozotocin-induced diabetic mouse models [18].

Concerted regulation of glucagon and insulin secretion plays a major role in the regulation of blood glucose. Hence, it is important to uncover and understand the interactions between α and β cells. However, islets are complex organisms containing multiple cell types and are consequently not an ideal experimental model to study α–β cell interactions. It is worth noting that the majority of pseudo islet research has been conducted solely on isolated β cell lines such as MIN-6. There has been a limited number of studies incorporating both α and β cells for cell interaction research [19,20,21,22,23], and most of these only demonstrate an impact on insulin secretion.

In the present investigation, we assessed the glucose-dependent changes of both insulin and glucagon secretion with MIN-6 cells and α-TC1-6 cells cultured as a mixed monolayer. Cell distribution was visualized by inverted microscopy and immunocytochemistry. To further clarify the mechanisms underlying various interactions, we examined the expression of selected genes related to auto/paracrine signaling and cell connection proteins.

## 2. Materials and Methods

### 2.1. Reagents and Buffers

All chemicals were purchased from Sigma-Aldrich Denmark A/S if not stated otherwise.

Modified Krebs–Ringer Buffer (M-KRB): 125 mM NaCl, 5.9 mM KCl, 1.2 mM MgCl_2_, 1.28 mM CaCl_2_, 5.0 mM NaHCO_3_, 25 mM HEPES, 0.015 mM BSA, and 1 mM glucose (pH 7.4).

SYTO 24 solution: 5 mM SYTO 24 green fluorescent nucleic acid stain (molecular probes; Invitrogen, Eugene, OR, USA) was mixed with sterile water to reach a final concentration of 10 μM. PBS: Dulbecco’s phosphate-buffered saline (Gibco, Paisley, UK).

Incubation buffer for antibody staining: 10% FBS and 0.5% Triton X-100.

### 2.2. Culture of α-TC1-6 and MIN-6 Cells

All procedures were carried out in a flow cabinet to ensure sterility. α-TC1-6 cells (a kind gift from Prof. Shimon Efrat, Tel Aviv University, Israel) between passages 36–48 were cultured in Dulbecco’s Modified Eagle’s Medium (DMEM; GIBCO-BRL, Paisley, UK), supplemented with 18 mM glucose, 10% FBS, and 100 units/mL penicillin in combination with 100 μg/mL streptomycin.

MIN-6 cells (generously donated by Dr. Yamamoto, Kumamoto University School of Medicine, Japan) between passages 42–51 were cultured in DMEM (GIBCO-BRL, Paisley, UK) , supplemented with 25 mM glucose, 10% FBS, 100 units/mL penicillin in combination with 100 μg/mL streptomycin.

Both cell types were incubated at 37 °C in a humidified atmosphere (5% CO_2_, 95% air) and passaged once a week. The medium was changed twice a week

### 2.3. Experimental Set-Up

To mimic the biological ratio of α and β cells in mouse islets, MIN6 cells and α-TC1-6 cells were mixed in the ratio of 3:1 as the “co-culture” condition. In total, 1 mL of the cell suspension was seeded in a 24-well Black Visiplate TC (Wallac Oy, Turku, Finland) as follows: (1) MIN-6, containing 2.625 × 10^5^ cells per well, (2) α-TC1-6 containing 0.875×10^5^ cells per well, (3) co-culture containing 2.625 × 10^5^ MIN-6 cells and 0.875 × 10^5^ α-TC1-6 cells per well. The cells were incubated in the medium with glucose levels of 5.5, 11.1, or 25 mM, respectively, for 72 h before an assay. See Table 1.

### 2.4. Glucagon Secretion and Insulin Secretion

After 72 h, cells were pre-incubated for 15 min in M-KRB (containing 1 mM glucose) (pH 7.4), followed by a 2 h incubation in M-KRB containing 1 mM (low) or 18 mM (high) glucose. Subsequently, 550 μL incubation medium was collected on ice, centrifuged for 2 min at 100× *g*, after which 200 μL supernatant was collected and kept at −20 °C for insulin and glucagon analysis.

Insulin was analyzed by radioimmunoassay using a guinea pig anti-porcine insulin antibody (Novo Nordisk, Bagsværd, Denmark). Mono-125I-(Tyr A14)-labelled human insulin was used as a tracer and rat insulin was used as a standard, both from Novo Nordisk. Ethanol was added to separate bound and free radioactivity. The inter- and intra-assay variation coefficients were both less than 5%. Glucagon analysis was performed using a radioimmunoassay kit (Millipore Research Park Drive, St Charles, MO, USA), according to manufacturer’s protocol. The glucagon antibody is specific for pancreatic glucagon and does not react with other islet polypeptides. The sensitivity limit for the assay is 20 pg/mL. We tested for cross interactions between the glucagon and insulin kits and found no cross bindings.

Following the secretion study, 20 μL SYTO 24 solution was added into each well and the plates were incubated for 40 min in a humidified atmosphere. The cell number was then measured and determined by FLUOstar Galaxy (BMG, Ramcon, Birkerød, Denmark). The emission and excitation were measured at 520 nm and 492 nm, respectively, and the gain was set at 40%.

### 2.5. Morphology and Immunocytochemistry

All procedures were performed at room temperature, unless otherwise noted. Cells were seeded on coverslips in accordance with the same experimental set-up described above, except that the 24-wells plates here are transparent Nunclon (Thermo Fisher Scientific, Roskilde, Denmark). After 72 h, morphology images of cells exposed to 25 mM glucose were taken under inverted optical microscopy (Olympus 50X, Barrington, IL, USA). Thereafter, cells were washed twice in PBS (phosphate buffered saline, pH 7.4), fixed in 4% paraformaldehyde for about 10 min, followed by two additional washes in PBS. The cells were permeabilized by incubation for 30 min in PBS containing 10% FBS and 0.5% Triton X-100 (Thermo Fisher Scientific). Primary antibodies including guinea pig anti-insulin (1:2000, DAKO, Cytomation A/S, Glostrup, Denmark) and rabbit anti-glucagon (1:1000, Thermo Fisher Scientific) were added to each well for overnight incubation. Afterwards, cells were washed 3 times with PBS and incubated with anti-rabbit Alexa 488 and anti-guinea pig Alexa 594 (1:1000, Thermo Fisher Scientific, Waltham, MA, USA) for 1 h prior to 2 washes in PBS. Finally, coverslips were mounted on object slides using a Fluoroshield Mounting medium with DAPI (Abcam, Cambridge, UK. Both primary and secondary controls were implemented to ensure the specific binding of the primary and secondary antibodies used in this assay. Following immunocytochemical staining, fluorescent images were obtained with the Leica application suite X. Images were acquired using a LEICA SP8 confocal system equipped with a white light laser, a 405 laser, and a 63x/1.2 objective using the following imaging settings: in between line scanning (1) Dapi: excitation 405 nm, detection 450–475 nm, (2) Alexa 488: excitation 488 nm, detection 505–535 nm, and (3) Alexa 594: excitation 488 nm, detection 600–650 nm.

### 2.6. Quantitative RT-PCR

#### 2.6.1. RNA Isolation

After 72 h incubation, total RNA of the cells from each well was isolated using the RNeasy^®^ minikit (Qiagen, Hilden, Germany) according to the protocol provided. RNA purity and concentration were evaluated by measuring absorbance at 260–280 nm (NanoDrop ND-8000 UV-vis spectrophotometer, Nanodrop Technologies, Wilmington, DE, USA). The 18S and 28S ribosomal bands were examined on a 0.7% nondenaturing agarose gel stained with SYBR green to evaluate the integrity of the RNA.

#### 2.6.2. Reverse Transcription Reaction

Quantitative real-time PCR was performed by BioXpedia A/S, Aarhus, Denmark. The samples were analyzed for the expression of 20 gene transcripts (Table 2) using mouse-specific TaqMan assays, as described previously [23]. One hundred nanograms of isolated RNAs were reversely transcribed into cDNA in a 20 μL final reaction volume using the TaqMan MicroRNA Reverse Transcription Kit. The reactions were performed according to the manufacturer’s protocol.

Reverse transcription was carried out using a 96-well Thermal Cycler (Veriti 96-well cycler, ThermoFisher, Waltham, MA, USA). The reaction conditions were: 25 °C for 10 min, 37 °C for 120 min, 85 °C for 5 s. All cDNA samples were kept at −20 °C until preamplification.

#### 2.6.3. Preamplification

A preamplification reaction was performed after the reverse transcription reaction on the cDNA samples using the TaqMan PreAmp Master Mix 2X (Thermo Fisher, Waltham, MA, USA) and a Primer Pool consisting of 0.2X (final concentration) of each TaqMan assay.

The reactions were performed in a 96-well cycler (Veriti 96-well cycler, Thermo Fisher, Waltham, MA, USA), according to the manufacturer’s protocol; enzyme activation at 95 °C for 600 s, followed by 14 cycles at 95 °C for 15 s and 60 °C for 240 s.

The amplified cDNA was analyzed on a 192.24 Dynamic Array on the Biomark HD Real-Time PCR system (Fluidigm, South San Francisco, CA, USA) using the standard Gene Expression Protocol: hold at 95 °C for 60 s (polymerase activation), followed by 35 cycles at 96 °C for 5 s (strand denaturation) and 60 °C for 60 s (annealing). All samples were measured in technical duplicates. The 2−∆∆Ct method was used to calculate the relative gene transcription. The transcription levels of the different gene transcripts were normalized against the mouse-specific TaqMan house-keeping gene assays, Table 2. Hprt1 (ABI, Mm00446968_m1) and β-actin (Mm02619580_g1) were used as house-keeping genes.

### 2.7. Viability of α-TC1-6 Cells

A total of 3 × 104 of α-TC1-6 cells were seeded in each well of a 96-well Visiplate TC (Wallac Oy, Turku, Finland). Cells were allowed to adhere overnight, then incubated with different concentrations of glucose (5.5, 18, or 30 mM, respectively). After 72 h, cell death rate in each well was calculated using a fluorometric assay kit base on the cell lysis and staining method (Cytotoxic Fluoro-test Wako) in FLUOstar Galaxy (BMG, Ramcon, Denmark).

### 2.8. Data and Statistical Analyses

We performed data analysis and graph design with GraphPad Prism 7.0 (GraphPad Software Inc., San Diego, CA, USA). Data are expressed as the mean ± standard error of mean (SEM). A two-tailed Student’s unpaired *t*-test was used to evaluate statistical comparisons between two groups. *p* < 0.05 was considered a significant difference. We have normalized insulin and glucagon secretion data on a per cell basis, and adjusted gene expression data according to the ratio of α and β cells (1:3).

## 3. Results

### 3.1. GSIS of Mono- and Co-Cultured MIN-6 and α-TC1-6 Cells

The effects on GSIS of mono- and co-cultured MIN-6 and α-TC1-6 cells are presented in Figure 1. After a 72 h incubation in 5.5 mM of glucose (A), co-cultured cells tended to secrete less insulin compared to mono MIN-6 cells when stimulated with either low or high glucose (*p* = 0.41 and 0.07). Mono MIN-6 cells and co-cultured cells showed no difference in GSIS after incubation in 11.1 (B) or 25 mM (C) glucose. Following a 72 h incubation in 5.5 mM and 11.1 mM glucose, high glucose tended to stimulate GSIS more than low glucose for both mono MIN-6 cells and co-cultured cells, even though significance was only observed in co-cultured cells incubated in 11.1 mM (*p* < 0.005). High glucose significantly prompted insulin release by approximately 2-fold compared to low glucose, both from mono MIN-6 and co-cultured cells after being incubated in 25 mM glucose for 72 h.

With high stimulation, the GSIS levels of both mono MIN-6 and co-cultured cells incubated in 25 mM glucose was higher than those incubated in 5.5 mM or 11.1 mM glucose (*p* < 0.0001) (Figure 1D,E). Meanwhile, there were no significant differences between the cells incubated in 5.5 mM and 11.1 mM glucose.

Mono-cultured α-TC1-6 cells secreted traces of insulin under all conditions.

### 3.2. Glucagon Secretion of Mono- and Co-Cultured MIN-6 and α-TC1-6 Cells

The glucose-dependent effects on glucagon secretion from mono- and co-cultured MIN-6 and α-TC1-6 cells are shown in Figure 2. After 72 h incubation in 5.5 mM glucose (A), co-culture of MIN-6 and α-TC1-6 cells tended to suppress glucagon secretion, albeit not significantly (*p* = 0.17 and 0.11), when stimulated with low and high glucose. Compared to mono α-TC1-6 cells, co-cultured cells demonstrated a clear decrease in glucagon after a 72 h exposure in 11.1 (B) or 25 mM (C) glucose, with an approximately 2-fold suppression in the latter (*p* < 0.01). Low or high glucose stimulation had no impact on glucagon secretion under any of the conditions (not significant).

When stimulated with high glucose, α-TC1-6 cells from 11.1 mM glucose secreted the most glucagon compared to cells from 5.5 (*p* < 0.05) and 25 mM (though not significant, *p* = 0.10) glucose (D). Furthermore, co-cultured cells from 25 mM glucose produced less glucagon compared to cells incubated in 5.5 or 11.1 mM glucose (E).

Glucagon secreted from mono-cultured MIN-6 cells could be observed under all conditions.

### 3.3. Distribution of MIN-6 and α-TC1-6 Cells in Co-Cultured System

Cell morphology images of mono MIN-6 cells, mono α-TC1-6 cells, and co-cultured cells are shown in Appendix A. After 72 h of exposure to 25 mM glucose, no obvious aggregates were formed in co-cultured cells compared to mono-cultured cells. In immunostainings, MIN-6 cells displayed mainly staining against insulin (green), together with detectable signals of glucagon (pink) in all three incubation conditions. Independently cultured α-TC1-6 cells presented only glucagon staining and co-cultured cells denoted both. In the co-culture system, α and β cells exhibited no specific distribution patterns or islet-like aggregates although clusters of α cells were found. Concurrently, signals of direct α–β contact were detected (see Figure 3).

### 3.4. Effects on Gene Expression in Mono- or Co-Cultured MIN-6 and α-TC1-6 Cells

To gain further insight into the effects of the co-culture of MIN-6 and α-TC1-6 cells, transcription levels of selected genes from each condition were examined, as shown in Figure 4. After a 72 h incubation in 11.1 or 25 mM glucose, RT-PCR analysis demonstrated that the co-culture of the two cell lines significantly decreased the gene expression of glucagon (Gcg) in comparison to mono-cultured α-TC1-6 (*p* < 0.05, Figure 4A), whereas no difference was shown when cells were incubated in 5.5 mM glucose. As depicted in Figure 4B–E, there was no apparent gene alteration found in co-cultured cells for the glucagon receptor (Gcgr), Glucokinase (GCK), GABA(A) receptor (Gabrg2), and E-cadherin (Cdh1). Moreover, co-culture did not change the insulin1 (Ins1) expression of MIN-6 cells (see Figure 4F). A similar pattern was seen for insulin2 (Ins2), except for cells co-cultured in the medium containing 25 mM glucose, where gene expression was upregulated (*p* < 0.01), as indicated in Figure 4G.

The gene expression of Insr slightly decreased in the co-culture condition as compared to both MIN-6 and α-TC1-6 cells, though it only became significant after incubation in 5.5 mM glucose (Figure 4H). Likewise, after co-incubation with α-TC1-6 cells for 72 h, connexin 36 (Gjd2) and Pancreatic and duodenal homeobox 1 (Pdx) expressed by MIN-6 were barely changed, except for a slight drop in cells grown in 5.5 mM glucose for both genes (*p* < 0.05), as revealed in Figure 4I,J. Lastly, Beta2/Neurod1, Akt1, N-CAM, PCSK2, PLCxd3, and Sirt 1 were expressed by both MIN-6 and α-TC1-6 cells (data not shown). Figure 4B–J are shown in Appendix A.

### 3.5. Impact of Different Concentrations of Glucose on the Viability of α-TC1-6 Cells

Figure 5 shows the viability of α-TC1-6 cells treated with 5.5, 18, or 30 mM glucose for 72 h. Cell death rate was significantly higher in 5.5 mM glucose than those in 18 or 30 mM (*p* < 0.05). No significant difference was found between the conditions in 18 and 30 mM.

## 4. Discussion

Pancreatic α–β cell interactions play a pivotal role in blood glucose and hormone regulation. It is of utmost importance to clarify the functional cross-talk between the two cell types to improve our understanding of diabetes. In the present study, we demonstrate that the co-culture of α and β cell lines does not affect insulin secretion but inhibits glucagon production. The expression of the glucagon gene, Gcg, is suppressed in co-cultured cells compared to mono-cultured α cell line. Instead of forming pseudo islets, we observed in the co-cultured conditions that α and β cells were irregularly distributed in a monolayer with noticeable compact spherical α cell clusters.

Pancreatic α cells secrete glucagon-related peptides such as glucagon, glucagon-like peptide-1 (GLP-1), and gastric inhibitory polypeptide. These peptides are critical for normal GSIS [24]. However, the recent results show that the presence of glucagon-secreting α-TC1-6 cells does not affect insulin secretion from MIN-6 cells when co-cultured in the ratio of 1:3; even a gentle decrease was seen in the 5.5 mM condition. This confirms the theory that α cells do not regulate insulin secretion through direct cell–cell contact. Insulin genes in mice comprise a two-gene system, preproinsulin 1 (Ins1) and preproinsulin 2 (Ins2). The two functional insulin genes code for exactly the same protein product. Gene expression of Ins1 was not altered in co-culture conditions (Figure 4F), which corroborates our findings in insulin secretion. For the gene expression of Ins2 (Figure 4G), only the co-cultured cells in 25 mM glucose experienced a minor elevation, which may, however, not be detectable at the protein level as no difference was shown in insulin release.

Our findings are supported by previous in vitro studies. Domenico and collaborators showed that insulin secretion is controlled by homologous but not heterologous cell–cell contact [25]. Brereton et al. co-cultured MIN-6 and αTC1 cells (also 3:1) and reported that the presence of α cells had no measurable effect on insulin secretory responses to nutrient or non-nutrient stimuli [19]. Moreover, Hamaguchi and co-workers demonstrated that insulin secretion from βTC cells may be suppressed due to direct contact with αTC cells when the mixing ratio of the two cell lines was 1:1 [21]. In another model, Kelly and colleagues demonstrated pseudo islets containing MIN-6, αTC1.9, and TGP52 cells that secreted similar amounts of insulin in comparison to that of “MIN-6 only” pseudo islets [22]. These results seem to show that endogenous glucagon has no effects on insulin secretion from β cells. Rodriguez-Diaz et al. confirmed this theory by showing that glucagon receptor antagonists did not affect insulin secretion from mouse islets. In contrast, they found in human islets that insulin secretion requires glucagon input from neighboring α cells [26].

Even though endogenous glucagon does not seem to be part of the paracrine β cell network, exogenous glucagon is capable of directly triggering insulin secretion [19,27,28,29,30]. Nonetheless, the mechanisms of paracrine glucagon altering insulin secretion remain inconclusive. Pancreatic β cells express glucagon receptors and GLP-1 receptors; Glucagon may stimulate insulin secretion through both receptors [31]. Additionally, Pipeleers and colleagues proposed that endogenously released glucagon increases cAMP levels in islet β cells and thus stimulates GSIS [32].

The present results show no difference in the gene expression of Gcgr (Figure 4B) between mono-cultured MIN-6 cells and co-cultured MIN-6/α-TC1-6 cells after incubation in 11.1 mM or 25 mM glucose for 72 h, which is inconsistent with the findings from the insulin secretion study.

Undisturbed insulin release in this study may indicate that glucagon enhances insulin secretion through a paracrine pathway rather than the direct contact-mediated communication of α cells with neighboring β cells. Alternatively, it may imply that α cells also have a feature that negatively regulates insulin secretion, forming a feedback loop with glucagon to maintain intra-islet insulin homeostasis. As mentioned, when increasing the proportion of α cells in the co-culture system, α cells do inhibit the insulin production of β cells [21]. The precise mechanism remains to be elucidated.

To the best of our knowledge, this is the first study to demonstrate that when stimulated with glucose, the presence of β cells inhibits glucagon secretion from α cells in a co-cultured system. After 3 days of incubation in different levels of glucose, glucagon release from α cells in the co-cultured conditions substantially decreased when stimulated with both 1 mM glucose and 18 mM glucose, even though the differences were not significant for cells grown in 5.5 mM glucose (Figure 2). Gene expression of glucagon, Gcg was significantly down-regulated after incubation in 11.1 or 25 mM glucose, but not in 5.5 mM (Figure 4A), which is in line with the findings from the glucagon secretion study.

Insulin is an essential mediator of the glucose-inhibited glucagon secretion. In the absence of insulin, glucose is not able to suppress glucagon release in vivo [33,34]. In fact, glucose may even stimulate glucagon secretion from α cells in the absence of β cells [35,36]. In mice, glucagon levels in the fed state are increased by the knock-out of α cell insulin receptors [30]. In the recent study, the gene expression of insulin receptor Insr from co-cultured conditions did not increase (Figure 4H), implying that insulin may not exert its effect (solely) through insulin receptors located on α cells. Vergari et al. suggested that insulin suppresses glucagon release by SGLT2-induced stimulation of somatostatin secretion [37]. Interestingly, without δ cells in the present co-culture system, glucagon secretion was still inhibited compared to mono-cultured α cells. This suggests the possibility of other signaling pathways for insulin to suppress glucagon secretion.

GABA is another paracrine factor released from β cells that might inhibit glucose-induced glucagon secretion from α cells [12,38]. Mouse GABA(A) receptors are expressed in pancreatic α cells, although mostly lacking in pancreatic β cells [39,40]. It is reported that both Intra-islet insulin and GABA may suppress glucagon release from α cells by activating GABA(A) receptors [12,41]. Activation of GABA(A) receptors in α cells leads to an influx of Cl^-^ and membrane hyperpolarization, sequentially resulting in the suppression of glucagon secretion. However, gene expression of GABA(A) receptors was not detected in α-TC1-6 cells in the present study (Figure 4D).

E-cadherin (Cdh1) is an important molecule in cell–cell adhesion, and connexin 36 (Gjd2) is the sole connexin expressed by pancreatic β cells. The co-culture of MIN-6 and α-TC1-6 cells shows no alteration on the gene expression of Cdh1 (Figure 4E), suggesting that cell–cell adherence remains the same with that in mono MIN-6 cells. Connexin 36 expression slightly decreased when co-cultured cells were incubated in 5.5 mM glucose (Figure 4I), which may imply that it plays a role in insulin secretion from β cells, since insulin levels were suppressed under this condition.

The glucose threshold for the suppression of glucagon is around 1 mM in rodent islets, while it is 5–7 mM for insulin stimulation [29,42,43,44,45]. This may explain the comparable stimulatory effects exerted by low or high levels (1 or 18 mM) of glucose. MIN-6 and α-TC1-6 are both immortalized cell lines that are typically cultured in 25 mM and 18 mM glucose, respectively. Thus, a glucose level of 5.5 mM may be considered as a “fasting state” for both cell types. In the cell viability experiment (Figure 5), we showed that in 5.5 mM of glucose, the death rate of α-TC1-6 cells was higher than those grown in 18 mM or 30 mM. This indicates that the glucose level of 5.5 mM for α-TC1-6 cells is a “harsh” environment where cells cannot grow or function properly.

Brereton et al. co-cultured MIN-6 and α-TC1-6 cells (3:1) for 6–8 days on gelatin-coated tissue culture flasks and observed the formation of three-dimensional islet-like clusters, i.e., the majority of α-TC1-6 cells were located in the periphery of the pseudo islets, while the MIN-6 cells formed in the core of the structure [19]. More recently, Kim et al. reported that co-cultured β TC-1 and α TC-6 cells (with the ratio of 1:1) formed aggregates while incubated in a gyratory shaker. In their system, α TC-6 became compact and spherical and β TC-1 cells did not show a specific pattern after 72 h incubation [46]. In this study, we mixed MIN-6 and α-TC1-6 cells at the ratio of 3:1 and maintained them in non-treated 24-well plates for 3 days. From both morphology (Appendix A) and immunostaining images (Figure 3), no islet-like cell clusters were observed. Early studies reported the formation of islet-like aggregates with or without shaking in a non-treated dish. However, Matta and team demonstrated that the rotational culture is more appropriate for pancreatic cells to reaggregate than the static culture [47,48]. Recently, Singh et al. found that sulfonylureas exert two opposite actions on glucagon secretion, i.e., a direct stimulation as in β cells, and an indirect inhibition via somatostatin signaling [49]. This corroborates the findings of Vergari et al. [37], showing that the therapeutic concentrations of insulin inhibit glucagon by an indirect (paracrine) mechanism mediated by intra-islet somatostatin release. However, the present study suggests other signal pathways independent of δ cells for insulin to suppress glucagon secretion. Unfortunately, we do not have data that point to specific alternative mechanisms of actions.

Our results confirmed this theory: Even without the formation of 3D architecture, the simplified monolayer co-culture system possesses some of the physiological characteristics of islets and could be an alternative to the study of islet function, thereby reducing animal use in research.

The limitations of this study are: (1) MIN-6 and α-TC1-6 cells are both clonal cells that may differ in function and behavior from normal islet cells in rodents. Whether the results from these experiments can be extrapolated to human physiology is unknown. Further studies with primary islet cell culture are needed to confirm if our results are translational. (2) In the co-culture system, the ratio of MIN-6 and α-TC1-6 cells may not remain at 3:1 after 3 days of growth. Though unlikely to change dramatically, it may lead to inaccurate gene expression results, since the data would have to be adjusted based on the ratio of 3:1. (3) The visualization of insulin is affected by the heterogeneity of insulin content, and thereby insulin staining of MIN-6 cells.

## 5. Conclusions

In a monolayer co-culture system, the presence of α cells does not change insulin secretion from β cells. In contrast, glucagon secretion is inhibited from α cells when co-cultured with β cells. When incubated in normal non-treated culturing plates, both α and β cells scatter randomly through the plates without the formation of pseudo islets. Further studies are warranted to reveal the molecular mechanism of glucagon suppression effects of β cells.

## Figures and Tables

**Figure 1 nutrients-13-02281-f001:**
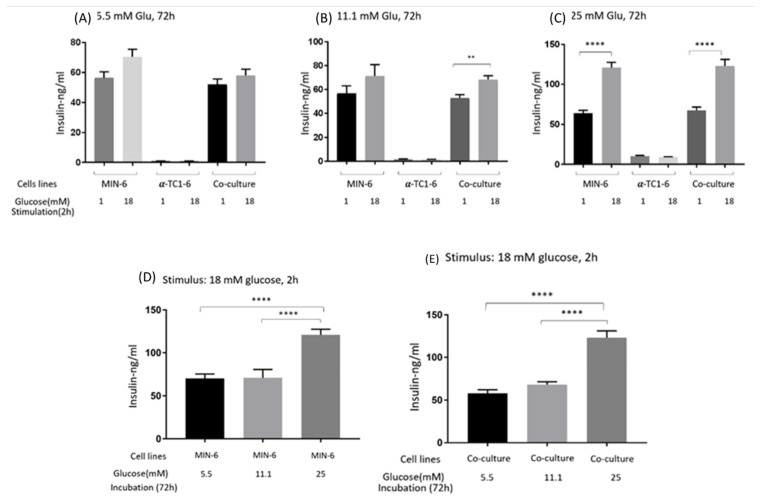
Insulin secretion from mono- or co-cultured MIN-6 and α-TC1-6 cells when stimulated with low (1 mM) or high (18 mM) glucose after 72 h incubation in (**A**) 5.5, (**B**) 11.1, or (**C**) 25 mM glucose, respectively. (**D**) Insulin secretion from MIN-6 cells when stimulated with high glucose after 72 h incubation in 5.5, 11.1, or 25 mM glucose, respectively. (**E**) Insulin secretion from co-cultured cells when stimulated with high glucose after 72 h incubation in 5.5, 11.1, or 25 mM glucose, respectively. Data are presented as the mean ± SEM of 18 samples per condition from three independent experiments. ** *p* < 0.01, **** *p* < 0.0001.

**Figure 2 nutrients-13-02281-f002:**
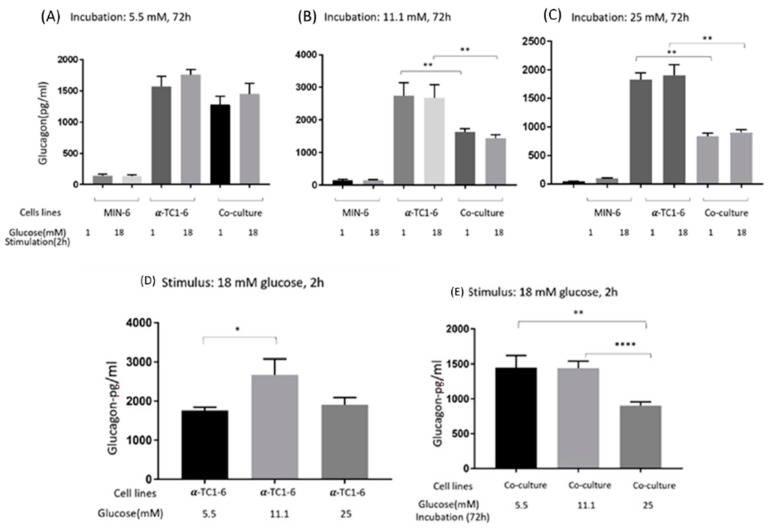
Glucagon secretion from mono- or co-cultured MIN-6 and α-TC1-6 cells when stimulated with low (1mM) or high (18 mM) glucose after 72 h incubation in (**A**) 5.5, (**B**) 11.1, or (**C**) 25 mM. (**D**) Glucagon secretion from α-TC1-6 cells when stimulated with high glucose after 72 h incubation in 5.5 mM, 11.1, or 25 mM glucose. (**E**) Glucagon secretion from co-cultured cells when stimulated with high (18 mM) glucose after 72h incubation in 5.5, 11.1, or 25 mM glucose. Data are presented as the mean ± SEM of 18 samples per condition from three independent experiments. * *p* < 0.05, ** *p* < 0.01, **** *p* < 0.0001.

**Figure 3 nutrients-13-02281-f003:**
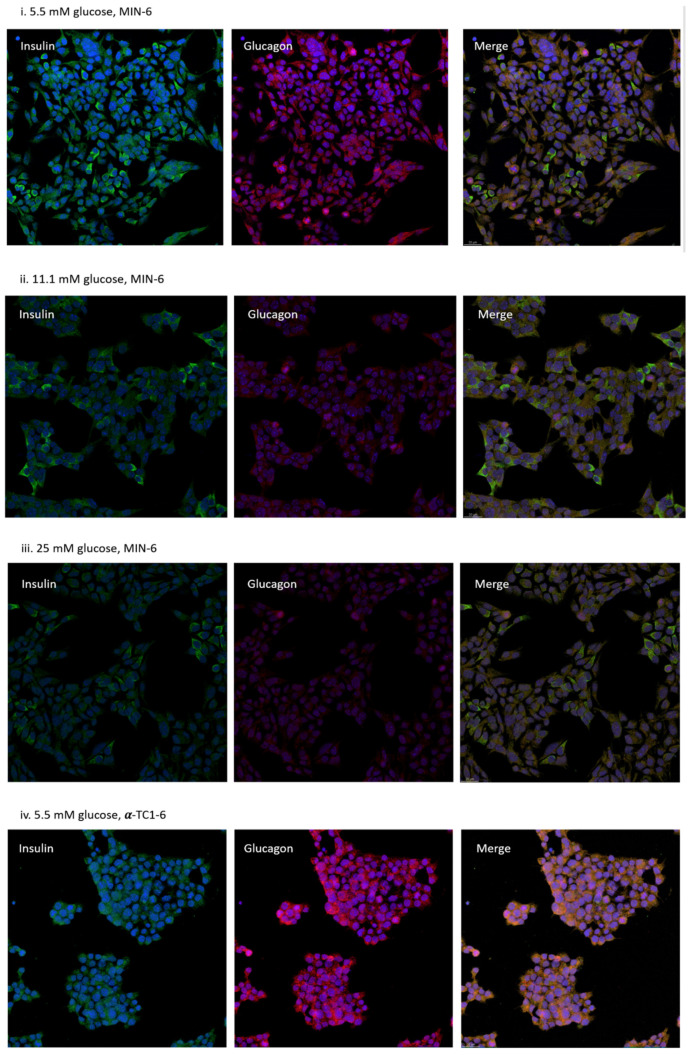
Representative immunocytochemistry microscopic appearance of mono-cultured MIN-6 cells (**i**–**iii**), mono-cultured α-TC1-6 cells (**iv**–**vi**), and co-cultured cells (**vii**–**ix**). Cells were visualized by laser scanning confocal microscopy after 72 h incubation in 5.5, 11.1, or 25 mM glucose. Cells were stained with both antibodies against insulin (green) and glucagon (Pink). In co-cultured conditions (**vii**–**ix**), MIN-6 cells were indicated by green arrows and α-TC1-6 cells were indicated by pink arrows. Scale bar are representative = 20 μm. Images of 1 representative view for each condition are shown (n = 5).

**Figure 4 nutrients-13-02281-f004:**
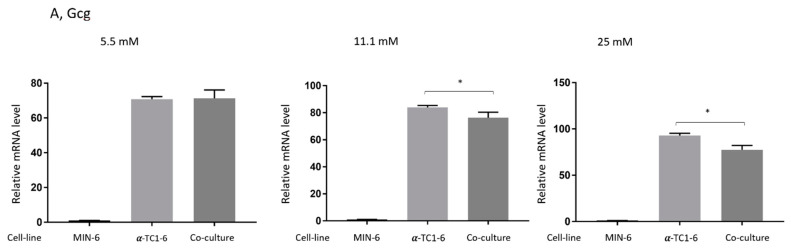
Effects on gene expression in mono- or co-cultured MIN-6 and α-TC1-6 cells after 72 h incubation in 5.5, 11.1, or 25 mM glucose. Transcript abundance of 20 specific genes was assessed by real-time RT-PCR using TaqMan assays; only the results of glucagon (Gcg) are shown here. Ten samples were obtained for each condition, and the samples were measured in quadruplicates. Data are presented as mean ± SEM. * *p* < 0.05, ** *p* < 0.01.

**Figure 5 nutrients-13-02281-f005:**
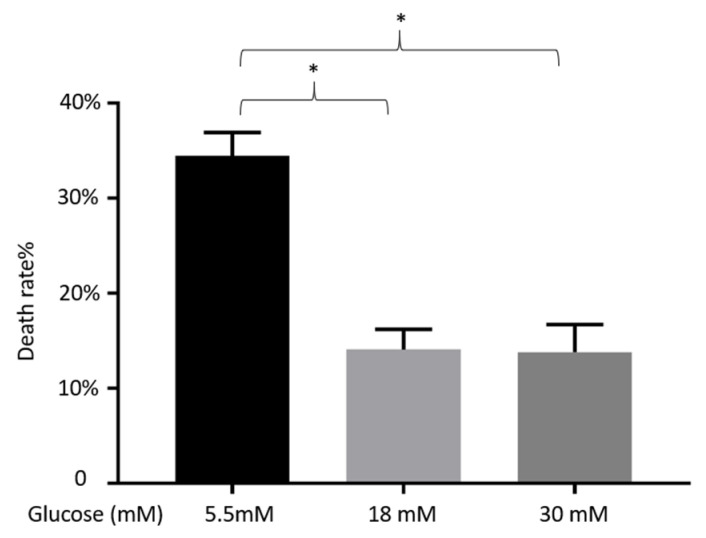
Effects on cell death rate of α-TC1-6 cells after 72 h incubation in 5.5, 18, or 30 mM glucose. Data are presented as the mean ± SEM of 24 samples per condition from three independent experiments. * *p* < 0.05.

**Table 1 nutrients-13-02281-t001:** Experimental set-up. MIN6 cells and α-TC1-6 cells were mono-cultured or co-cultured in the medium containing 5.5, 11.1, or 25 mM glucose, respectively.

Glucose Levels	Cell Types	Cell Numbers/Well
5.5, 11.1, or 25 mM	MIN6 cells	2.625 × 10^5^ MIN6 cells
	α-TC1-6 cells	0.875 × 10^5^ α-TC1-6
	Co-culture	2.625 × 10^5^ MIN6 cells + 0.875 × 10^5^ α-TC1-6

**Table 2 nutrients-13-02281-t002:** List of genes examined for expression from mono- or co-cultured MIN-6 and α-TC1-6 cells.

Gene Symbol	TaqMan Assay ID	Name
Ins1	Mm01259683_g1	Insulin I
Ins2	Mm00731595_gH	Insulin II
Pdx 1	Mm00435565_m1	Pancreatic and duodenal homeobox 1
Akt1	Mm01331626_m1	Thymoma viral proto-oncogene 1
GCK	Mm00439129_m1	Glucokinase
PLCxd3	Mm01307828_m1	Phosphatidylinositol-specific phospholipase C, X domain containing 3
Sirt 1	Mm01168521_m1	Sirtuin 1
Beta2/Neurod1	Mm01946604_s1	Neurogenic differentiation 1
Lcn2	Mm01324470_m1	Lipocalin 2
Insr	Mm01211875_m1	Insulin receptor
Gcg	Mm00801712_m1	Glucagon
Gcgr	Mm00433546_m1	Glucagon receptor
PCSK2	Mm00500981_m1	Pyruvate dehydrogenase kinase, isoenzyme 1
Gabrg2	Mm00433489_m1	Proprotein convertase subtilisin/kexin type 2
Gjd2	Mm00439121_m1	Gamma-aminobutyric acid (GABA) A receptor
Gja1	Mm00439105_m1	Connexin36
Cdh1	Mm01247357_m1	Connexin43, gap junction protein
N-CAM	Mm00493049_m1	E-cad, Ecad, L-CAM
Slc2a2	Mm00446229_m1	L1 cell adhesion molecule
Slc39a5	Mm00511105_m1	Solute carrier family 2 (facilitated glucose transporter), member
		Solute carrier family 39 (metal ion transporter)

## Data Availability

Can be deliver by contacting the authors.

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
