# Peer review of "Pancreatic β Cells Inhibit Glucagon Secretion from α Cells: An In Vitro Demonstration of α–β Cell Interaction"

_nutrients, 2021, doi:10.3390/nu13072281_

Round 1

Reviewer 1 Report

In this interesting study by Wenqian Gu and coworkers aimed to ascertain the impact of ? and ? cell co-culture on hormone secretion. Specifically, the authors demonstrated that pancreatic β-cells suppress glucagon secretion by co-culture of α-β cell interaction apparently via direct cell to cell contact.

The manuscript is well designed and straight forward, is written clearly, the conclusions and the hypothesis are supported by the results and are coherent with the current bibliography. Indeed, this manuscript is original and very well-reasoned and for this reason the article is easy to follow for both experts in the field and a wide audience.

Minor remarks:

Although the inhibition of glucagon secretion has been extensively studied in previous models, the co-culture conditions seem like an interesting study to cover the interactions between α and β cell in vitro on hormone secretion. In fact, recent studies showed that the inhibition of glucagon secretion is mediated by somatostatin signaling (Nat Commun. 2019 Jan 11;10(1):139) and that δ-cells inhibit α-cells more efficiently than β-cells (Mol Metab. 2021 Jun 9;101268). However, the present study suggests other signals pathways independent to δ-cells for insulin to suppress glucagon secretion. The authors may speculate a bit more in detail in the discussion, on these other possible vias independent to δ-cells to suppress glucagon release.

However, despite the authors suggest that the translationality of the results may be limited, due to the use of clonal cells, something that perhaps could also be improved with the use of primary cell culture, the contribution of these findings and the impact of these results on the pathophysiology of diabetes should be discussed further.

Author Response

Answers to reviewers:

Reviewer 1:

Comments and Suggestions for Authors

In this interesting study by Wenqian Gu and coworkers aimed to ascertain the impact of ? and ? cell co-culture on hormone secretion. Specifically, the authors demonstrated that pancreatic β-cells suppress glucagon secretion by co-culture of α-β cell interaction apparently via direct cell to cell contact.

The manuscript is well designed and straight forward, is written clearly, the conclusions and the hypothesis are supported by the results and are coherent with the current bibliography. Indeed, this manuscript is original and very well-reasoned and for this reason the article is easy to follow for both experts in the field and a wide audience.

Minor remarks:

Although the inhibition of glucagon secretion has been extensively studied in previous models, the co-culture conditions seem like an interesting study to cover the interactions between α and β cell in vitro on hormone secretion. In fact, recent studies showed that the inhibition of glucagon secretion is mediated by somatostatin signaling (Nat Commun. 2019 Jan 11;10(1):139) and that δ-cells inhibit α-cells more efficiently than β-cells (Mol Metab. 2021 Jun 9;101268). However, the present study suggests other signals pathways independent to δ-cells for insulin to suppress glucagon secretion. The authors may speculate a bit more in detail in the discussion, on these other possible vias independent to δ-cells to suppress glucagon release.

Answer: We thank for your valuable comments. We have now underlined that other possible vias independent of the δ-cells to suppress glucagon release. However, we do not have data on specific alternative and therefor have avoided pure speculations. We have now added the following sentences on this issue in the discussion page 15 lines 445-52 : “Recently, Singh et al. found that sulfonylureas exert two opposite actions on glucagon secretion i.e. a direct stimulation as in β-cells and an indirect inhibition via somatostatin signaling [49]. This corroborates the findings of Vergari et al [50] showing that therapeutic concentrations of insulin inhibit glucagon by an indirect (paracrine) mechanism mediated by intra-islet somatostatin release. However, the present study suggests other signalling pathways independent of δ-cells for insulin to suppress glucagon secretion. Unfortunately, we do not have data that point to specific alternative mechanisms of actions.” 

  1. Singh B, Khattab F, Chae H, Desmet L, Herrera PL, Gilon P. KATPchannel blockers control glucagon secretion by distinct mechanisms: a direct stimulation of α-cells involving a [Ca2+]crise and an indirect inhibition mediated by somatostatin. Mol Metab. 2021 Jun 9:101268. doi: 10.1016/j.molmet.2021.101268. Epub ahead of print. 

50 Vergari E, Knudsen JG, Ramracheya R, Salehi A, Zhang Q, Adam J, Asterholm IW, Benrick A, Briant LJB, Chibalina MV, Gribble FM, Hamilton A, Hastoy B, Reimann F, Rorsman NJG, Spiliotis II, Tarasov A, Wu Y, Ashcroft FM, Rorsman P. Insulin inhibits glucagon release by SGLT2-induced stimulation of somatostatin secretion. Nat Commun. 2019 Jan 11;10(1):139. doi: 10.1038/s41467-018-08193-8. 

However, despite the authors suggest that the translationality of the results may be limited, due to the use of clonal cells, something that perhaps could also be improved with the use of primary cell culture, the contribution of these findings and the impact of these results on the pathophysiology of diabetes should be discussed further.

Answer: Thank you for the input. We agree that the translationality could be improved with the use of primary cell culture and will consider this for future studies. This is now mentioned in the discussion at Page 16 lines 461-2: ” Further studies with primary islet cell culture to emphasize if our results are translational”

Reviewer 2 Report

In the paper entitled “Pancreatic beta-cells inhibits glucagon secretion from alpha-cells: an in vitro demonstration of alpha-beta cell interaction” Gu and co-workers investigated the glucose dependent changes in glucagon and/or insulin secretion of MIN-6/alpha TC1-6 co-culture monolayer.

Authors demonstrated that glucagon secretion due to alpha cells is negatively affected by beta cells in a of MIN-6/alpha-TC1-6 monolayer mixed co-culture.

The paper is well written, the rationale is clear and figures are exhaustive. Nevertheless, despite the high level of this study, the scientific soundness of this paper should be improved. Authors should deeply analyze the biological significance of their finding particularly the glucagon inhibition on pancreatic alpha cells.

Some concerns are to be assessed to the authors:

1-In Material and methods section:

- Table 2 (row 188) the gene name associated with Taqman Assay ID Mm00433546_m1 (Gcgr-glucagon receptor) does not fit with name associated with. Please check and correct.

- Row 189 Authors assessed alpha-TC1-6 cell viability; however no data have been provided concerning MIN-6 cells viability. Authors should investigate and reveal it.

- MIN6 are generally culture using β-mercaptoethanol. Have been the authors used this compound?

- Why did authors use 18 mM of glucose for insulin secretion? Several papers reported 20-25 mM is suitable for this procedure.

2- Results section:

- Authors showed insulin secretion from both of MIN-6, alfa-TC1-6 and in co-culture condition: in graph marked with “A”, Authors should check statistical analyses: differences  between insulin secretion in MIN-6 after 1mM glucose and  18 mM glucose treatment seem to be statistically significant.

- In the immunofluorescence analysis Authors assessed that alpha cells alone showed only glucagon staining. It is right only for 25 mM glucose stimulation. Moreover, Authors highlighted the co-localization of insulin and glucagon suggesting alpha and beta contact. However, glucagon is also present in MIN6 alone and the co-colture staining at 5.5 mM of glucose that may influence the co-localization signal.

- Authors reported a significant decrease in both glucagon release and mRNA at 25 mM of glucose stimulation. Thus, how can they justify the increased fluorescence signal in the same experimental conditions?

- MIN6 showed a slight secretion of glucagon and its presence is also confirmed by immunofluorescence analysis. However no mRNA is detected. Can the author explain this discrepancy?

- Have Authors also tested shorter time for insulin secretion (15, 30 and 60 minutes)? 

- Authors should investigate and reveal the signaling pathway underlying glucagon release inhibition in MIN-6/alpha-TC1-6 monolayer mixed co-culture displayed in figure 2.

- Immunochemistry microscope analysis should be improved overall in co-culture images; beside insulin and glucagon staining, Authors should stain MIN-6/alpha-TC1-6 monolayer mixed co-culture with a suitable antibody capable to discriminate MIN-6 from alpha-TC1-6 in the same microscopy field. Otherwise, it will be very difficult recognizing co-culture experimental conditions.

- Authors should confirm real time data displayed in figure 4 by using western blot analysis or other technique able to confirm that mRNA variation is actually associated to protein variation. At least the genes that are statistically significant modulated.  

Minor concerns

1- In Material and methods section row 105, 106 Scientific notation should be reported as apex.

Author Response

Reviewer 2:

Comments and Suggestions for Authors

In the paper entitled “Pancreatic beta-cells inhibits glucagon secretion from alpha-cells: an in vitro demonstration of alpha-beta cell interaction” Gu and co-workers investigated the glucose dependent changes in glucagon and/or insulin secretion of MIN-6/alpha TC1-6 co-culture monolayer.

Authors demonstrated that glucagon secretion due to alpha cells is negatively affected by beta cells in a of MIN-6/alpha-TC1-6 monolayer mixed co-culture.

The paper is well written, the rationale is clear and figures are exhaustive. Nevertheless, despite the high level of this study, the scientific soundness of this paper should be improved. Authors should deeply analyze the biological significance of their finding particularly the glucagon inhibition on pancreatic alpha cells.

Some concerns are to be assessed to the authors:

1-In Material and methods section:

- Table 2 (row 188) the gene name associated with Taqman Assay ID Mm00433546_m1 (Gcgr-glucagon receptor) does not fit with name associated with. Please check and correct.

Answer: Thank you for this relevant point. We have now corrected the gene name associated with Taqman Assay ID Mm00433546_m1 (Gcgr-glucagon receptor) by adding glucagon receptor. Please see Page 5 Line 190

- Row 189 Authors assessed alpha-TC1-6 cell viability; however no data have been provided concerning MIN-6 cells viability. Authors should investigate and reveal it.

Answer: Thank you for the suggestion. Unfortunately, we have no data on the viability of MIN-6 cells, as our mean focus has been on the alpha-TC1-6 cell.

- MIN6 are generally culture using β-mercaptoethanol. Have been the authors used this compound?

Answer: Thank you for the question. No we did not use β-mercaptoethanol in the present studies.

- Why did authors use 18 mM of glucose for insulin secretion? Several papers reported 20-25 mM is suitable for this procedure.

Answer: Thank you for the input. Several papers have used 16.7 mM glucose for insulin secretion – and we have previously applied 18 mM glucose in a number of studies. We do not think that results would have differed if we have used 20 mM glucose. The glucose threshold for suppression of glucagon is around 1 mM in rodent islets, while the glucose threshold for stimulation of insulin is 5-7 mM.

2- Results section:

- Authors showed insulin secretion from both of MIN-6, alfa-TC1-6 and in co-culture condition: in graph marked with “A”, Authors should check statistical analyses: differences  between insulin secretion in MIN-6 after 1mM glucose and  18 mM glucose treatment seem to be statistically significant.

Answer:  Thank you for the point. We have checked again and there is unfortunately no significant different, even there is a very strong tendency.

- In the immunofluorescence analysis Authors assessed that alpha cells alone showed only glucagon staining. It is right only for 25 mM glucose stimulation. Moreover, Authors highlighted the co-localization of insulin and glucagon suggesting alpha and beta contact. However, glucagon is also present in MIN6 alone and the co-culture staining at 5.5 mM of glucose that may influence the co-localization signal.

Answer: Thank you for this interesting point. Yes, it correct that MIN-6 cells monoculture also secrete small amount of glucagon. We also found that mono-cultured ?-TC1-6 cells secrete trace of insulin under all conditions. We have already described this at page 7 line 237. We have tried to test for cross interaction between the kits, and found no cross binding between glucagon and insulin. We have added this to the method section: Page 3 L 125 “ We tested cross interaction between the glucagon and insulin kits and found no cross bindings”.

- Authors reported a significant decrease in both glucagon release and mRNA at 25 mM of glucose stimulation. Thus, how can they justify the increased fluorescence signal in the same experimental conditions?

Answer:  Thank you for the point. We did not conclude an increased fluorescence signal at 25 mM glucose stimulation. We have not applied immunocytochemistry as a quantitative measurement. The images we have chosen were only representatives to illustrate that glucagon and insulin are present.

- MIN6 showed a slight secretion of glucagon and its presence is also confirmed by immunofluorescence analysis. However no mRNA is detected. Can the author explain this discrepancy?

Answer: Good point.  However, as seen at figure 4 A, the glucagon secreted from the MIN6  is at very low levels only. .

- Have Authors also tested shorter time for insulin secretion (15, 30 and 60 minutes)?

Answer: Thank you for these good ideas. However, due to the substantial work in this experimental setup we only tested for insulin secretion at the time points indicated, however, this could be an interesting suggestion for the next study.  

- Authors should investigate and reveal the signaling pathway underlying glucagon release inhibition in MIN-6/alpha-TC1-6 monolayer mixed co-culture displayed in figure 2.

Answer: Thank you very much for the good suggestion. As this is the first kind of experiments with this type of clonal cell types, we would be interested to investigate further on  the signaling pathway underlying inhibition of glucagon release inhibition in MIN-6/alpha-TC1-6 monolayer mixed co-culture in future studies.

- Immunochemistry microscope analysis should be improved overall in co-culture images; beside insulin and glucagon staining, Authors should stain MIN-6/alpha-TC1-6 monolayer mixed co-culture with a suitable antibody capable to discriminate MIN-6 from alpha-TC1-6 in the same microscopy field. Otherwise, it will be very difficult recognizing co-culture experimental conditions.

Answer: Thank you very much for the good suggestion. You are totally right. We will take this with us in our future studies.

- Authors should confirm real time data displayed in figure 4 by using western blot analysis or other technique able to confirm that mRNA variation is actually associated to protein variation. At least the genes that are statistically significant modulated. 

Answer: This is a good point which we will take into account for the next study to be performed. As we already have substantial  amount of data in this paper, we will include this technology e.g.  western blot analysis in the next paper.

Minor concerns

1- In Material and methods section row 105, 106 Scientific notation should be reported as apex.

Answer: Thank you. You are totally right. This have now been added to Page 3 line 106-7